# Can the New and Old Drugs Exert an Immunomodulatory Effect in Acute Myeloid Leukemia?

**DOI:** 10.3390/cancers13164121

**Published:** 2021-08-16

**Authors:** Francesco Tarantini, Cosimo Cumbo, Luisa Anelli, Antonella Zagaria, Giorgina Specchia, Pellegrino Musto, Francesco Albano

**Affiliations:** 1Hematology and Stem Cell Transplantation Unit, Department of Emergency and Organ Transplantation (D.E.T.O.), University of Bari “Aldo Moro”, 70124 Bari, Italy; f.tarantini@studenti.uniba.it (F.T.); cosimo.cumbo@uniba.it (C.C.); luisa.anelli@uniba.it (L.A.); antonella.zagaria@uniba.it (A.Z.); pellegrino.musto@uniba.it (P.M.); 2School of Medicine, University of Bari “Aldo Moro”, 70124 Bari, Italy; giorgina.specchia@uniba.it

**Keywords:** acute myeloid leukemia, AML treatment, immunomodulation

## Abstract

**Simple Summary:**

The advent of novel immunotherapeutic strategies has revealed the importance of immune dysregulation and of a tolerogenic microenvironment for acute myeloid leukemia (AML) fitness. We reviewed the “off-target” effects on the immune system of different drugs used in the treatment of AML to explore the advantages of this unexpected interaction.

**Abstract:**

Acute myeloid leukemia (AML) is considered an immune-suppressive neoplasm capable of evading immune surveillance through cellular and environmental players. Increasing knowledge of the immune system (IS) status at diagnosis seems to suggest ever more attention of the crosstalk between the leukemic clone and its immunologic counterpart. During the last years, the advent of novel immunotherapeutic strategies has revealed the importance of immune dysregulation and suppression for leukemia fitness. Considering all these premises, we reviewed the “off-target” effects on the IS of different drugs used in the treatment of AML, focusing on the main advantages of this interaction. The data reported support the idea that a successful therapeutic strategy should consider tailored approaches for performing leukemia eradication by both direct blasts killing and the engagement of the IS.

## 1. Introduction

Acute myeloid leukemia (AML) is a heterogeneous hematological malignancy characterized by an abnormal blast proliferation, variable prognosis and response to treatment, and high mortality [1]. AML is considered an immune-suppressive neoplasm that adopts protective strategies against the immune system (IS), including the downregulation of major histocompatibility complex (MHC) class I and class II expression, deregulation of dendritic cells (DCs) function, expansion of T-regulatory (T-reg) cells, upregulation of antiapoptotic mechanisms, loss of tumor antigen expression and secretion of immunosuppressive cytokines [2,3,4,5]. The common features of IS dysregulation in de novo AML are the impaired activity of natural killer (NK) cells and macrophages and T-cell exhaustion and the inability to form immunological synapses with blasts [5]. Moreover, there is evidence of a disruption of the immune checkpoint mechanisms in AML, with the upregulation of PD-L1 and other negative checkpoint molecules, such as CTLA-4 and LAG-3 [6]. Therefore, it is widely accepted that, in AML, immune evasion involves both cellular and environmental players [7,8], even if a better understanding of their precise contribution to leukemia immune tolerance and response to treatment is required [7,8,9].

The advent of immune checkpoint inhibitors (ICI) and chimeric antigen receptor-T (CAR-T) cell approaches revealed the importance of immune dysregulation and suppression in the onset of leukemia, encouraging strategies to engage the IS via novel immunotherapeutic approaches [10]. Although the actions of these new agents on leukemic cells has been amply described, their effects on the leukemia microenvironment and the IS are not yet completely understood.

In this review, we report the possible effects of the drugs approved for AML treatment on the IS (summarized in Table 1), focusing on the main advantages of these interactions. We intentionally refrain from discussing bispecific T-cell engager antibody, ICI and CAR-T cell approaches, because the latter have been conceived to unleash the patient’s immune system to fight leukemic cells, and this kind of therapy should be addressed separately.

## 2. Conventional Chemotherapy

Despite the introduction of new drugs, the backbone of induction therapy in AML is still chemotherapy, the “7+3” regimen being the most widely used [13]. In the past decade, various studies have demonstrated that cytotoxic drugs are associated with a perturbation of both B- and T-cell-mediated immune responses in AML [40,41]. Furthermore, it has been shown that cytotoxic drugs could exert a direct immunogenic effect against tumor cells, mediating their release of tumor-associated antigens and damage-associated molecular pattern molecules (DAMPS), which can stimulate the innate immune response to trigger the so-called “immunogenic cell death” (ICD). Upon treatment, the tumor cells may alter their interactions with the host IS [11,42,43,44,45]. Notably, DAMPS are associated with many biological processes and diseases; their release from the cells and tissues is observed during inflammation, sepsis, fungal infections, cancer and cell death by apoptosis, ferroptosis and NETosis. Moreover, live cells are capable of exporting DAMPS on their surface under stress conditions. [12,46]. A coordinated emission of DAMPS characterizes ICD in the context of cancers; in the early phase, the chaperone calreticulin (CRT) and heat shock protein-70 and -90 (HSP-70 and HSP-90) translocate on the cell surface, initiating the apoptotic process and stimulating DC maturation; in the late phase, the nonhistone chromatin-binding protein high-mobility box group-1 (HMGB1) and the adenosine triphosphate (ATP) molecules are released in the extracellular milieu, thereby allowing the complete maturation and functionality of DCs; these latter shift cytokine production towards an immunostimulatory profile [44,47].

Among the different classes of cytotoxic drugs used in AML, anthracyclines (i.e., daunorubicin) have been demonstrated to activate, in vitro, an IS-mediated cell death through the translocation of CRT on the neoplastic cell surface (Figure 1) [48]. Moreover, cultured AML cells co-express CRT, HSP-70 and HSP-90 on their surfaces [48]. Interestingly, in vivo CRT exposure on AML blasts has been related to an enhanced anticancer immune response and a better prognosis [49,50,51]. A closer analysis in the murine model of AML cells expressing CRT showed a robust immune-mediated control of leukemia, T cells and DCs, dependent upon the host expression of type I interferon (IFN-1) [52].

In line with the high incidence of relapse of AML after conventional chemotherapy-based induction therapy [13,53], both the in vivo and in vitro data showed, upon treatment with daunorubicin, an immunosuppressive response alongside the above-mentioned effects [47]. Specifically, daunorubicin has been related to the release of ATP from AML blasts, correlated, in turn, with an increase of the T-regs and, notably, with an enhanced tolerogenic pathway activation (mediated by DCs), ultimately affecting the T cells [47]. Of note, daunorubicin may have a further role in the interplay between IS and malignancies. In fact, when conjugated with alpha-fetoprotein and tested in a murine model of cancer, it recognized and selectively eliminated myeloid-derived suppressor cells (MDSCs), which are thought to be at least partly responsible for the failure of anticancer immunity [54].

Less evidence is available in regard to cytarabine. The in vitro data demonstrated that cytarabine at low concentrations has, on one hand, as expected, a high antiproliferative effect against AML blasts, and, on the other, it may decrease T-cell function, alter the expression of DAMPs and reduce the release of HSP-90 [40]. Interestingly, experimental data demonstrated that the resistance to cytarabine in T-cell leukemia cell lines may cause the selection of clones with an increased sensitivity to natural killer (NK)-mediated cell lysis, concretely associated with a higher surface expression of ligands for the NK cell-activating receptor NKG2D [55]. However, these findings have not yet been explored in the context of AML. No data are available as regards the new injectable liposomal formulation of cytarabine and daunorubicin, CPX-351. Given its pharmacokinetic and pharmacodynamic features [15] and the previous knowledge, studies exploring the interplay between CPX-351 and the IS would be of great interest.

These first observations about cytotoxic agents allowed us to highlight that two drugs, daunorubicin and cytarabine, members of the same category, can exert an opposite immunomodulatory effect. This difference should be considered when they are combined with new drugs, whose mode of action will be discussed below.

DAMP: damage-associated molecular pattern molecules, NK: natural killer, HMAs: hypomethylating agents, LAAs: leukemia-associated antigens, HLA: human leukocyte antigens, DC: dendritic cells, T-reg: T-regulatory cells, IS: the immune system.

## 3. FLT3 Inhibitors

FMS-like tyrosine kinase 3 (FLT3) is a type III receptor tyrosine kinase usually expressed on hematopoietic progenitors; its activation upon binding with the FLT3 ligand induces multiple intracellular signaling pathways, leading to cell proliferation, differentiation and survival [16]. *FLT3* mutations occur in approximately 30% of AML patients [17]; the most frequent alterations are internal tandem duplications (ITD) in the juxtamembrane domain and point mutations of the D835 hotspot in the tyrosine kinase domain (TKD). Both cases result in the constitutive activation of FLT3 signaling and a consequent uncontrolled cell proliferation [16].

Given the frequency and the strong association of *FLT3* mutations with a poor prognosis in AML, it has been the focus of studies of new drugs in the last two decades. First-generation drugs (midostaurin, sorafenib, lestaurtinib and tandutinib) are multi-kinase inhibitors conceived to target FLT3 regardless of the mutational subtype. Second-generation inhibitors (gilteritinib, quizartinib and crenolanib) were designed to be highly selective and sensitive, targeting mutated FLT3 molecules [16].

The immunomodulation role of these drugs in AML treatment is less well-known, the inhibitor most closely studied being sorafenib. In light of the results of the SORMAIN trial in terms of relapse-free survival (RFS) and overall survival (OS) [56], sorafenib is a consistent option for maintenance therapy in the post-transplant AML setting; the anti-relapse effect may be partly due to off-target sorafenib-mediated immunomodulation. In fact, there is evidence that, in AML patients relapsing after an allogeneic hematopoietic stem cell transplant (allo-HSCT), the administration of sorafenib induces an increased bone marrow (BM) infiltrate of T CD8+CD279+ “exhausted” lymphocytes [57]. Furthermore, an altered expression in the BM of the genes involved in proinflammatory, immune responses and angiogenesis (*fibroblast growth factor 1 (FGF1), toll-like receptor 9 (TLR9), collagen type 4 alpha 3 chain (COL4A3), interleukin-12 (IL-12), nitric oxide synthase 2 (NOS2), colony stimulating factor 2 (CSF2)* and *angiopoietin-like 4 (ANGPTL4)*) is observed as an effect of the sorafenib off-target inhibition of the RAF/MEK/ERK and PI3K/mTOR/AKT pathways [57]. Moreover, the clinical features of graft-versus-host disease (GVHD) indirectly confirm an underlying immune activation [57]. Coherently, it has been shown that sorafenib may indirectly regulate the activation of CD8+ T-lymphocytes and natural killer cells (NKs) by restoring the production of interleukin-15 (IL-15) in *FLT3*-ITD-mutated clones (Figure 1) [14]. In the leukemic clone, sorafenib activity may also be extended to an increase of IFN-γ production, which may reverse the downregulation of MHC Class II, known to be involved in post-allo-HSCT relapse [18]. The data regarding sorafenib activity on T-regs are conflicting. On one hand, a reduction of the T-regs proportions in the peripheral blood (PB) of AML patients has been demonstrated [57]; on the other, in a comparative in vitro analysis among sorafenib, tandutinib, quizartinib and midostaurin, only this latter was related to a reduction of the T-regs pool in PB samples from both healthy and AML patients, maybe due to the inhibition of the off-target kinases (KIT, PDGFR, SRC and VEGFR) [58]. After the results of the RATIFY trial, midostaurin was FDA- and EMA-approved for use in combination with conventional chemotherapy in induction, consolidation and maintenance therapy for *FLT3*-mutated AML [59]. A post-transplant use is, therefore, feasible, as it may enforce the graft versus leukemia (GvL) effect, considering its activity on T-regs, while not altering the T-cell reactivity [60].

Lastly, similar results have recently been reported for gilteritinib, a dual FLT3/AXL inhibitor (with a minimal activity on KIT) approved for the use in relapsed/refractory *FLT3*-mutated AML. It has been tested in the post-transplant maintenance setting, yielding encouraging results [24]. Notably, as observed with sorafenib, during gilteritinib treatment, an increase in the total number of NKs and CD8+ T cells has been reported.

## 4. Hypomethylating Agents

Epigenetic changes due to aberrant methylation patterns are relatively common in AML, leading to either local or global effects on the gene expression [59]. Hypomethylating agents (HMAs) can induce transient DNA hypomethylation and the consequent re-expression of suppressed genes [19]. Differently from myelodysplastic syndrome (MDS), their efficacy in AML is independent of the mutational status of epigenetic modifiers such as *IDH1, IDH2, DNMT3A* and *TET2* [61,62]. Currently, decitabine (5-aza-2′-deoxycitidine) and azacytidine (5-azacitidine) are approved for the treatment of AML [63]. There is evidence that HMAs can influence IS actors at various stages, even if the studies have been conducted mainly in the context of MDS.

Azacytidine use has been associated with an upregulation of CD40 (acquired during DC maturation) and CD86 (a costimulatory molecule able to activate T cells) on the DC surfaces and a T-helper 17 polarization in advanced MDS and AML (Figure 1) [20]. Moreover, decitabine has been shown to enhance immune activation upon donor lymphocyte infusion (DLI) in the post-allo-HSCT AML relapse setting by activating DCs [64]. Conversely, a reduction of MDSCs upon the treatment with HMAs has been demonstrated in tumor-bearing mice [25].

In detail, as regards T cells, multiple observations, sometimes conflictual, in association with HMAs have been made. The expression of *PD-1* (an inhibitory receptor of T cells) is epigenetically regulated [21], and an altered methylation status has been found in hematological malignancies after treatments with HMAs [26]. The *PD-1* upregulation consequent to demethylation has been observed in AML upon treatment with decitabine [27]. Its ligands, PD-L1 and PD-L2, were also upregulated and associated with the resistance to HMAs [27]. This finding offers a rationale for the combined use of ICI targeting the PD-1/PD-L 1 axis and with HMAs, in line with the available clinical data [65].

Furthermore, there is evidence that HMAs may alter T-cell priming and activation by regulating the expression of costimulatory molecules such as CD80 and CD86 [20,66]. Recent studies have shown that a sequential treatment with decitabine and ICI is associated with a reversion of the exhaustion-associated de novo methylation programs in CD8+ T cells [67], confirming a synergistic therapeutic effect.

Aside from their qualitative effects, HMAs exert a quantitative effect on the subsets of T cells in MDS and AML, where the azacytidine treatment has been shown to reduce T-helper 1, T-helper 2 and T-helper 17 cells [68,69]. Further evidence has demonstrated an expansion of T-regs in MDS, with a high risk of progression to AML [70]. Both azacytidine and decitabine treatments may influence the T-reg numbers and plasticity, determining a functional shift consisting of an immunosuppressive loss effect and the acquisition of T-effector features [68,71]. This is consistent with multiple observations in the context of post-allo-HSCT-relapsed AML [72,73,74,75].

Furthermore, the HMA effects in AML and MDS may extend to an augmented T-cell recognition of CD34+ blasts and, at least transiently, the expression of leukemia-associated antigens (i.e., SART-3, MAGE-A1, MAGE-A2, TAG-1, NY-ESO-1, NUF3, GnTV, CDCA1, Sp17 and WT1) and HLA antigens (Figure 1) [73,76,77,78,79,80]. In addition, a study of the effects of decitabine on the highly cytotoxic γδ T-lymphocytes subset showed that decitabine could inhibit γδ T-cell proliferation and activity [81].

Regarding NK functionality, despite several in vitro and in vivo studies, there is no consensus on the effects of HMAs; both molecules seem to exert different effects [22,23,28,82,83]. Nevertheless, treatment with decitabine has been associated with an increased expression of the NK-activating receptor ligand NKG2DL and augmented antibody-dependent cellular cytotoxicity (ADCC) in post-therapy BM samples from AML patients [84]. Moreover, data from the in vivo and in vitro AML models exposed to a combination of HMAs and adoptive NKs demonstrated that not only were NK functions not impaired but, also, that their antileukemic activity was boosted upon exposure to decitabine. Additionally, treatment with decitabine resulted in increased numbers of NK in the BM compartment, suggesting that decitabine may positively modulate NK activity, trafficking and tumor targeting [29].

## 5. IDH Inhibitors

Isocitrate dehydrogenases (IDH) are metabolic regulatory enzymes that convert isocitrate to α-ketoglutarate (α-KG) in the tricarboxylic acid cycle. IDH-1 is the cytoplasmic enzymatic isoform, while IDH-2 is located in the mitochondrion [85]. *IDH* mutations occur in 20% of AML patients (mainly *IDH1*-R132, *IDH2*-R140 and *IDH2*-R172) [85]. Mutated IDH leads to the accumulation of oncometabolite R-2-hydroxyglutarate (R-2-HG) that ultimately blocks cell differentiation, promotes leukemogenesis by altering the histone and DNA methylation and confers genomic instability [85]. The IDH1 and IDH2 inhibitors, ivosidenib and enasidenib, respectively, have been approved for use in AML [86].

Although little evidence is available about the interplay between the immune system and IDH in AML, some observations have been made following extensive studies conducted on gliomas, the most common form of central nervous system neoplasm in which IDH gene mutations are recurrent [87,88,89]. Firstly, it was shown that as IDH-mutant glioma cells tend to export R-2-HG to protect themselves from its toxic effects, the extracellular concentration of R-2-HG is found to be five-fold higher than the intracellular concentration [87,90]. Microenvironmental antitumoral T-lymphocytes are capable of importing R-2-HG through specific receptors (SLC13A3 and SLC22A6). Once inside, R-2-HG impairs the transcriptional activity of the nuclear factor of activated T cells (NFAT), thus suppressing T-cell activity [87]. Interestingly, the inhibition of SLC13A3 improved T-cell proliferation [87]. Moreover, further studies have shown that IDH mutant gliomas exhibit lower numbers of CD3+ and CD8+ lymphocytes in the tumor milieu because of the reduced T-cell expression, attracting the chemokines CXCL9 and CXCL10 and their regulator, STAT1 [88]. In murine glioma models, the effect of IDH mutations has been shown to extend to altered neutrophil functions, complement-mediated phagocytosis suppression and NK ligand reduction [89,91].

In the context of IDH-mutated AML, the R-2-HG accumulation seems to disturb the IS on the metabolic front through the destabilization of hypoxia-inducible factor-1 (HIF-1a), thereby determining a shift from aerobic glycolysis to oxidative phosphorylation [92]. Consequently, the differentiation and function of T cells is altered, increasing the T-reg pool and reducing the T-helper 17 polarization [91]. In light of these data, the immunosurveillance might be impaired by the presence of IDH-mutant clones in AML. Even if no data are yet available, it can be assumed that targeted therapies may, realistically, be able to restore the lost equilibrium.

## 6. BCL-2 Inhibitors

Apoptosis is a critical process for a wide range of tissue functions and for maintaining homeostasis, which involves the control of immunity [93]. Proapoptotic and antiapoptotic molecules comprising the BCL-2 family, whose factors are critical regulators of the intrinsic apoptotic pathway, have a role in hematological malignancies [30]. Notably, in AML, *BCL-2* overexpression has been associated with enhanced cell survival, apoptosis evasion and resistance to therapy [31,32,94]. Venetoclax is an oral BH3 mimetic that is highly selective for BCL-2 inhibition and approved for use in combination with HMAs or low-dose cytarabine to treat older or unfit adults in standard chemotherapy for newly diagnosed AML [95].

Targeting BCL-2 determines the “off-target” effects, including impairment of the homeostasis of immune cells of the B, T and myeloid lineages (Figure 1) [96]. There is little evidence regarding venetoclax-mediated immune modulation in AML. Nevertheless, data from Chronic Lymphocytic Leukemia (CLL) patients demonstrated that venetoclax-based treatment did not only lead to hematological responses but also decreased the absolute number of nonmalignant B, T and NK cells [97]. Furthermore, the simultaneous decline of tumor-supportive and immune-suppressive T cells (namely T-follicular helper, T-regs and PD1+CD8+ T cells), and the restoration of NK cell functions, were observed [97]. This observation is in contrast with the previous experimental data underlining the sensitivity of both normal and leukemic B cells, but not T cells, to venetoclax activity [98]. This may suggest that the effects on the T-cell repertoire observed in CLL may be a consequence of a perturbation of cancer and IS crosstalking rather than a direct pharmacological action [97]. In the murine model, venetoclax treatment has been associated with a reduced number of total T cells; curiously, their functions were not altered. The observed reduction was attributed specifically to naïve T cells, while the T effectors were treatment-insensitive [99]. These data were confirmed upon the administration of venetoclax in healthy human subjects, showing an increased proportion of T-effector cells following treatment [99]. The latest evidence revealed an unexpected augmented cytotoxic activity of CD3+CD4-CD8- double-negative T cells (DNTs) and CD8+ lymphocytes upon treatment with venetoclax and azacytidine in de novo AML [100]. On the one hand, the blasts are primed for T-cell killing by an azacytidine-induced viral mimicry response; on the other hand, venetoclax moderately increases the reactive oxygen species (ROS) in T cells, leading to augmented effector activity without affecting the T-cell viability. These findings suggest a possible explanation of the efficacy of venetoclax in the context of untreated AML, where the IS is still qualitatively and quantitatively unaffected by chemotherapy. As the spectrum of indications for venetoclax-based therapies in hematology is constantly expanding, further studies are needed to clarify whether the data described above are a direct drug effect or, intriguingly, the mirror of its action on malignant clones and the consequent disruption of the interplay between tumoral cells and the IS.

## 7. Hedgehog Pathway Inhibitors

The Hedgehog (HH) family of intercellular signaling proteins is of crucial importance in embryonic development, tissue polarity control and stem cell maintenance [101]. Aberrations of the HH pathway are involved in tumorigenesis [33]. In AML, it is fundamental for leukemic stem cell homeostasis and is implicated in the acquisition of drug resistance [102], as an upregulation of the HH pathway components has been observed in vitro in chemo-resistant AML cell lines [34]. In normal tissue, transmembrane protein smoothening (SMO) is essential for the HH pathway activation [35,103] and is involved in maintaining leukemic stem cells dormancy through its effector, GLI2 [104]. HH signaling can be aberrantly activated in AML through the epigenetic silencing of the GLI3 pathway suppressor [104].

The oral SMO inhibitor glasdegib is the only FDA-approved drug for use in combination with low-dose cytarabine to treat newly diagnosed AML patients over 75 years old or who are not candidates for intensive induction chemotherapy [102]. Its activity may disrupt the leukemic stem cell dormancy, thus causing re-entry into the cell cycle [104]. To date, there are no specific data in regard to the glasdegib-related immune effects in AML. Nevertheless, the consistent evidence collected in solid cancers may open new perspectives for the combined pharmacological strategies involving HH inhibitors [33].

Aberrant HH signaling can induce an immunosuppressive microenvironment through several mechanisms. In basal cell carcinoma, experimental data showed that tumoral keratinocytes secrete transforming growth factor-beta (TGFβ), which can reduce the number of tumor-infiltrating lymphocytes and enhance the recruitment of MDSCs in mice bone marrow [36,105].

Further data from the breast cancer xenograft model confirmed, on one hand, the MDSC-related observation but, on the other, showed a reduction of MDSCs, T-regs and M2 macrophages upon treatment with vismodegib, the other HH inhibitor [106].

Moreover, involvement of the HH effector protein GLI2 in T-cell polarization and CD4+ cell differentiation in immunosuppressive T-regs has been demonstrated [37,107,108]. Interestingly, the experimental data on gastric cancer showed that GLI2 can drive the overexpression of *PD-L1*; the use of anti-PD-L1 antibodies in this context triggered the apoptosis of tumor cells [109]. Data from basal cell carcinoma and medulloblastoma with HH aberrations confirmed the upregulation of PD-L1, suggesting a regulatory function of HH in controlling the expression of immune checkpoint molecules [110,111]. Lastly, the treatment with HH inhibitors in basal cell carcinoma led to a restored expression of MHC-I (whose downregulation is a known immune escape mechanism) [112] and increase of the environmental T cells [113]. These data seem to establish an immune-modulatory activity of the HH pathway. The combination of azacytidine and glasdegib has a synergistic activity against AML [102]; this effect probably relies on the HMA-mediated reactivation of GLI3 [102]. Nevertheless, considering the above observations about the IS, further studies aiming to address the effects of this combination on the AML immune milieu are warranted.

These data underline the importance of ICI (PD-1/PD-1L axis) in the context of HH aberrations; therefore, combinations of the ICI and HH inhibitors in the treatment of AML could be considered for further investigation.

## 8. p53 Targeted Therapies

Often defined as “the guardian of the genome” [114], *TP53* has been widely studied in light of its multiple functions and dynamics, being involved in cell fate decisions in response to DNA damage and the consequent modulation of the cell cycle, senescence and apoptosis [115,116]. The somatic mutations and genomic loss of *TP53* due to deletions of the 17p chromosome occur in 5–10% of de novo AML cases [117] and in 40% of patients with adverse-risk cytogenetics and treatment-related myeloid neoplasms [118] and are associated with poor prognosis [119]. *TP53* aberrations have been shown to modulate both the immune and inflammatory responses in malignancies [120], affecting the T-cell differentiation [121,122], T-regs recruitment [123], PD-1 and PD-L1 expression [124,125] and STAT3 signaling [126]. Moreover, the restoration of the *TP53* functions could promote innate and adaptive immune responses and increase the immunogenicity of the tumor cells [126].

Regarding AML and MDS harboring *TP53* mutations, the newest data showed an evasive immune phenotype characterized by the overexpression of PD-L1, associated with miR-34a expression [127], reduced numbers of both cytotoxic and helper T cells and the expansion of T-regs and MDSCs [38]. Accordingly, a further in vitro analysis of AML with a loss-of-function mutation of *TP53* revealed a high expression of interferon-gamma and inflammation pathway genes [128].

Therapeutic approaches aiming to stabilize both the wild-type and mutant p53 are still investigational, so their contextual immunological effects are as-yet unexplored. APR-246 is a prodrug that covalently binds to the mutant p53 core domain, promoting apoptotic mechanisms and cell cycle arrest pathways [129,130]. Its trials have yielded encouraging results both as a single agent and in combined regimens with azacytidine and venetoclax [39,131].

Interestingly, the preclinical data from the melanoma mice model suggest a synergistic action of APR-246 with ICI, resulting in a decreased proliferation and increased cytolytic activity of CD8+ T cells, which was not observed with ICI alone. These findings have been further confirmed in solid cancer patients [132].

p53 transcriptional activity is physiologically regulated by the E3 ubiquitin-protein ligase MDM2; therefore, MDM2 inhibitors are currently being tested in AML [133], suggesting idasanutlin to be the most selective and potent drug. Encouraging results are being observed with idasanutlin both as a single agent and associated with conventional chemotherapy or venetoclax [134]. An in vivo analysis of the influence of idasanutlin on the differentiation and inflammatory responses of human macrophages demonstrated no significant effects; conversely, idasanutlin could attenuate the macrophages’ ability to respond to proinflammatory stimuli during monocyte-to-macrophage differentiation [135].

A new oral anti-MDM-2, DS-5272, has been tested in a mouse AML model driven by MLL-AF9 and patient-derived xenograft models of human AML [136]. Notably, its antileukemic effect was attenuated in immunodeficient mice and immunocompetent mice with NK cell depletion, thereby indicating NK, but not T cells, as critical mediators of the drug therapeutic efficacy [136]. The same study pointed out that DS-5272 triggered an immunoinflammatory response characterized by PD-L1 overexpression (Figure 1), in line with the above observations. These data could further suggest a synergistic combination of *TP53*-targeted therapies with ICI.

Considering that the results of the clinical trials involving p53-targeted therapies [137] in AML are recent or still ongoing, a paucity of precise data regarding drug-related IS effects in this pathological context is comprehensible. Nevertheless, the seminal data reported here encourage new attempts to focus on the possible interplays between p53-directed pharmacological strategies and the immunological environment in order to individuate the best synergistic approaches.

## 9. Discussion

The surveillance by a host’s IS is critical in the control of cancer. The term “immunoediting” was first adopted 20 years ago to explain how a cancer and IS interact in three phases: elimination, in which innate and adaptive immune cells recognize and clear transformed cells; equilibrium, characterized by a high genetic instability in the resistant neoplastic cells with the aim to evade from IS control; escape: the variant clones expand by downregulating (editing) immunogenic molecules and adopting IS-suppressing mechanisms. Therefore, genetic instability and mutations are thought to be the drivers of cancer progression [5]. Notably, in the Cancer Genome Atlas Research Network, AML has been found to have one of the lowest mutational burdens, with an average of 13 mutations in the genes of de novo AML patients [138]. Moreover, AML has been demonstrated to present among the lowest levels of neoantigens, whose expression is associated with a higher IS-mediated cytolytic activity [5]. Therefore, AML is not surprisingly considered a low immunogenic tumor.

In line with this concept, the experimental data on *MLL-ENL*-driven AML have shown that immunoediting has a limited role in leukemic proliferation [139]. Moreover, in AML harboring the *AML1/ETO* fusion gene, immune escape could even be inhibited by the upregulation of CD48 on the cell surface, thereby allowing NK recognition and killing [140]. This provides a further rationale for the good prognosis associated with this AML subtype. However, some of the typical AML mutations are associated with the expression of neoantigens on the cell surface, stimulating the IS; this is the case for *FLT3-ITD* and *NPM1* [5]. Moreover, mutations in *IDH1/2* are associated with impairment of the T-cell functions, as discussed above.

Otherwise, the success of allogeneic hematopoietic stem cell transplantation (allo-HSCT) in AML has demonstrated the efficacy of the immune system in eradicating leukemic cells; the antileukemic response in allo-HSCT is, in fact, mediated by the graft versus leukemia (GvL) effect; donor T CD4+, CD8+ and NK cells recognize and clear residual blasts on the basis of immunological nonidentity, which relies mainly on the cell surface expression of both major and minor histocompatibility complex molecules and leukemia-associated proteins such as Wilms tumor 1, proteinase 3, survivine and immature laminin receptor [141]. Furthermore, the successful use of Donor lymphocytes infusion (DLI) in the context of post-allo-HSCT relapse is proof of the importance of GvL; strategies aiming to maximize the GvL effect without raising the graft-versus-host disease (GVHD) risk are ongoing and include combinational approaches with DLI and HMAs or ICIs and the use of fractionated DLI [142].

The ever-expanding armamentarium of therapeutic agents and the refinement of consolidated procedures, such as the aforementioned allo-HSCT, have led to a considerable improvement in the outcome of AML patients. Nevertheless, the RFS and OS rates are still far from satisfactory. The success of allo-HSCT should be seen as two faces of the same coin, illustrating, on one hand, the power and variability of the treatments aiming to eradicate the leukemic clones and, on the other, the efficacy of the GvL immunosurveillance. This concept supports the idea that a successful therapeutic strategy should consider an integrated approach, simultaneously arming both weapons: the antileukemic and the IS-stimulating effects. Consequently, many drugs, both investigational and approved in AML, target or harness the IS: ICI, monoclonal antibodies, bispecific T-cell engagers (BiTEs) and CAR-T cells. It should be noted that, differently from the encouraging results obtained in non-Hodgkin’s lymphoma and acute lymphoblastic leukemia, CAR-T cells struggle to reach success in AML; the data show that unsatisfactory responses may be partly due to the hostile AML microenvironment, where the structural components of the bone marrow niche, soluble factors and suppressive immune cells such as T-regs and myeloid-derived suppressor cells (MDSC) are capable of globally reducing the CAR-T efficacy [143]. Aside from this, the data from combined strategies are encouraging, with HMA-based regimens topping the list. Considering all these premises, we reviewed the main effects of different AML-directed drugs on the IS. As expected, the HMAs emerged as the most commonly explored class, capable of inducing changes in the IS and conferring immunogenicity to AML blasts. Epigenetic regulation involves all cells, and its perturbation upon treatments with HMAs may explain the broad spectrum of “off-target” effects observed, including the IS-related ones. Therefore, the drugs’ off-target effects seem to offer a clue. Notably, we reported that the FLT3 inhibitor effects on the IS are mainly associated with inhibiting off-target kinases.

Similarly, the impairment of apoptosis through BCL-2 inhibition does not only exert its effects on the leukemic clone but also affects the homeostasis of the immune cell. The BCL-2 inhibitor effects on the IS suggest another possible explanation, in line with the data from the drugs inhibiting IDH and HH and with TP53-targeted agents: the eradication of leukemic cells upon the treatment might ultimately lead to the disruption of a tolerogenic microenvironment. Moreover, daunorubicin effects reinforce this concept, since triggering immunogenic cell death is a critical factor for a successful response to chemotherapy.

## 10. Conclusions

All these considerations highlight the crucial role of the IS in the context of AML therapeutic success. An increasing knowledge of these cellular mechanisms may support the design of tailored therapeutic approaches performing clone eradication by direct blast killing and engagement of the IS. The identification of the biomarkers of IS activation upon drug exposure, such as an increase of CD8+ and NK after treatment with gilteritinib and DNTs after venetoclax/HMA, would provide further information on the IS impact on the prognosis and survival of AML patients.

## Figures and Tables

**Figure 1 cancers-13-04121-f001:**
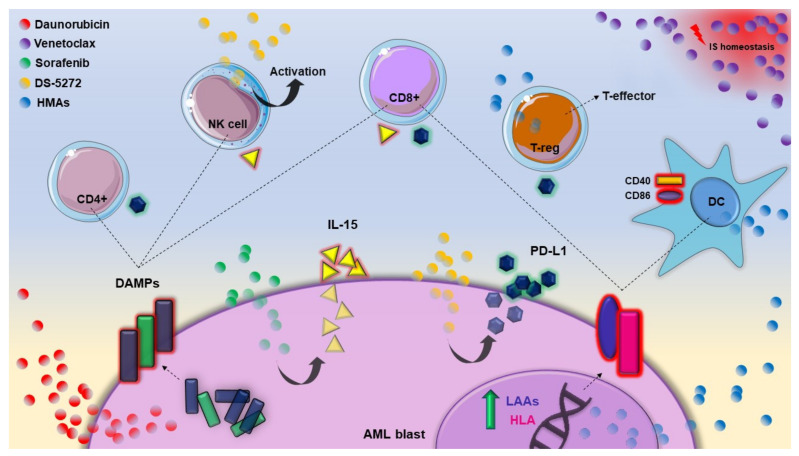
Crosstalk between the leukemic cell and its immunologic counterpart. The main molecular effects of the drugs used for AML treatment. From left to right: daunorubicin induces IS-mediated neoplastic cell death through the translocation of DAMPs on the cell surface. Sorafenib activates CD8+ T-lymphocytes and NKs by restoring the blast production of IL-15. DS-5272 activates NKs and triggers an immunoinflammatory response through *PD-L1* overexpression. HMAs increase the T-cell recognition of leukemic cells through the overexpression of LAAs and HLA, activate DCs through the overproduction of CD40 and CD86 and promote a T-reg shift to a T-effector. Venetoclax impairs the IS homeostasis.

**Table 1 cancers-13-04121-t001:** Main immunomodulatory effects of the AML therapeutic approaches. The principal effects on the IS of the drugs currently used in AML patient management. For each molecular mechanism, the main literature references are reported. AML: acute myeloid leukemia, IS: immune system, DAMP: damage-associated molecular pattern molecules, GvL: graft versus leukemia, NK: natural killer cell, DC: dendritic cell, T-reg: T-regulatory cells, R-2-HG: R-2-hydroxyglutarate, MDSC: myeloid-derived suppressor cell and MHC-I: major histocompatibility complex class I.

Therapeutic Approach	Activity	Drug	Immunomodulation	References
Conventional chemotherapy	Cytotoxicity	Daunorubicin	IS mediated blast death, through DAMPs translocation on cell surface	[11,12]
		Cytarabine	T cell reduction and DAMP expression alterations	[13]
FLT3 inhibitors	Kinase inhibition	Midostaurin	GvL effect reinforced	[14]
		Sorafenib	CD8+CD279+ T lymphocytes increasing Proinflammatory, IS and angiogenic genes regulation CD8+ T lymphocytes and NK activation	[15,16,17]
		Gilteritinib	NK and CD8+ T cells increased	[18]
Hypomethylating agents	Gene expression activation	Decitabine	DC activation (*CD40* and *CD86* overexpression) T cells inhibition (PD-1/PD-l1 axis activation) T helper 1, 2, 17 reduction T-reg shift to T-effector Increasing T cells recognition of CD34+ cells NK activation (*NKG2DL* and *ADCC* overexpression)	[19,20,21,22,23]
		Azacytidine		[24,25,26,27]
IDH inhibitors	IDH inhibition	Ivosidenib	No available data in AML Target therapies could restore the IS equilibrium altered by R-2-HG accumulation (assumption)	[28,29]
		Enasidenib		
BCL-2 inhibitors	BH3 mimetic	Venetoclax	Homeostasis impairment of B, T and myeloid immune cells	[30,31,32]
Hedgehog pathway inhibitors	Cell cycle re-entry	Glasdegib	No available data in AML MDSC recruitment, restored MHC-I expression, T cells increase (evidence from solid cancers)	[33,34,35,36,37]
p53 targeted therapies	Cell cycle arrest and apoptosis	APR-246	No available data in AML Increased cytolytic activity of CD8+ T cells (evidence from solid cancers)	[38]
	MDM2 inhibition	DS-5272	NK activation and blast *PD-L1* overexpression	[39]

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
