# Peer review of "Can the New and Old Drugs Exert an Immunomodulatory Effect in Acute Myeloid Leukemia?"

_cancers, 2021, doi:10.3390/cancers13164121_

Round 1

Reviewer 1 Report

This review focuses on drugs used for treating AML and their effect on host immune cells, which brings immune system into the AML treatment.

The review was well-written and interesting. The information and comments provided in this review should help to bring clinicians and immunologists together to better target AML.

Adding information on the effect of the AML drugs on adoptively transferred immune cells will further increase the quality of this review.   

This review discussed the effect of anti-AML drugs on patient immune system.  Adoptive T cell therapies, including donor lymphocyte infusion and CAR-T therapy, have been in clinical trials to treat AML patients. It would be interesting to include some information regarding the effect of these AML drugs on survival, persistence, memory phenotype and overall anti-leukemia activity of the adoptively transferred cells.

Some minor comments:

  1. Several recent publications reported that Venetoclax enhances T cell mediated cytotoxicity against AML both in vitro and in vivo. It would be good to include in the review.
  2. Fig.1: Please label the big cell as AML. Delete the sentence “Data deriving from other hematological malignancies or solid tumors are not shown in this image.”
  3. Line 187, delete CLA-4 as CTLA-4 is not a PD-1 ligand.
  4. Line 258, add word inhibition after BCL-2
  5. Line 303: “environmental lymphocytes” should be changed to “tumour infiltralting lymophocytes”

Author Response

Reviewer #1:

“…Adoptive T cell therapies, including donor lymphocyte infusion and CAR-T therapy, have been in clinical trials to treat AML patients. It would be interesting to include some information regarding the effect of these AML drugs on survival, persistence, memory phenotype and overall anti-leukemia activity of the adoptively transferred cells...”

According to the Reviewer’s suggestion, these topics have been discussed (page 10, lines 465-480).

“…Several recent publications reported that Venetoclax enhances T cell mediated cytotoxicity against AML both in vitro and in vivo. It would be good to include in the review…”

The text has been modified as suggested (page 7, line 323).

“…Fig. 1: Please label the big cell as AML. Delete the sentence “Data deriving from other hematological malignancies or solid tumors are not shown in this image…”

“…Line 187, delete CTLA-4 as CTLA-4 is not a PD-1 ligand…”

“…Line 258, add word inhibition after BCL-2…”

“…Line 303: “environmental lymphocytes” should be changed to “tumor infiltrating lymphocytes…”

Accordingly, the text has been modified.

Reviewer 2 Report

The authors reviewed the “off-target” effects on the IS of different drugs used in the treatment of AML, focusing on the main advantages of this interaction. However, the following issues should be considered:

  1. The authors should discuss the effect of innate and adaptive immune cells onto the leukemic cells. The important role of the immune responses warrants a brief summary of the current theories.
  2. There is a positive correlation of lymphocyte recovery after chemotherapy and low relapse rate. By which way do the authors separate the anti-leukemia drug immune effect from the endogenous immune responses after bone marrow recovery? In rare cases, a spontaneous remission of AML patients has been associated with activation of the immune system through pathogen infections. The authors should make a major effort to analyze clearly the results of the studies in order to support that the immunologic alterations are an effect of the anti-leukemic drug and not the results of endogenous immune response, immune response to infections or simply a recovery of the immune system after a complete remission.
  3. The authors stated that conventional chemotherapy exerts a direct effect against tumor cells, mediated through DAMPs. DAMPs have been shown to regulate inflammation in fungal diseases, as well. The authors should describe the specific DAMPs signaling in AML patients, supporting that this is drug and not pathogen related.
  4. The role of genetic mutations in leukemia immunosurveillance should also be discussed.
  5. The authors should analyze which of the immunologic effect of the anti-leukemia drug might be a biomarker of prognosis or a target for immunotherapy.
  6. The mechanisms of GVL effect should be analyzed and correlated with the anti-leukemic drug-associated deregulation of the immune system.
  7. The term “cancer immunoediting” should be described.

Author Response

Reviewer #2:

“…The authors should discuss the effect of innate and adaptive immune cells onto the leukemic cells. The important role of the immune responses warrants a brief summary of the current theories…”

According to the Reviewer’s suggestion, this topic has been now added to the introduction (page 1, lines 32-41)

“…There is a positive correlation of lymphocyte recovery after chemotherapy and low relapse rate. By which way do the authors separate the anti-leukemia drug immune effect from the endogenous immune responses after bone marrow recovery? In rare cases, a spontaneous remission of AML patients has been associated with activation of the immune system through pathogen infections. The authors should make a major effort to analyze clearly the results of the studies in order to support that the immunologic alterations are an effect of the antileukemic drug and not the results of endogenous immune response, immune response to infections or simply a recovery of the immune system after a complete remission...”

We thank the Reviewer for suggesting a topic of such great interest. Nevertheless, we think that the point that he raised could be the object of a comprehensive review, given the importance and the amplitude of the topic. The aim of our paper, otherwise, is to collect all observations available regarding the immune system status in AML after exposure to different drugs. Therefore, we could not discuss the suggested topic.

“…The authors stated that conventional chemotherapy exerts a direct effect against tumor cells, mediated through DAMPs. DAMPs have been shown to regulate inflammation in fungal diseases, as well. The authors should describe the specific DAMPs signaling in AML patients supporting that this is drug and not pathogen related...”

Accordingly, we discussed the ubiquity of DAMPs exposure and the current hypothesis regarding the immunogenic cell death mechanisms in cancer and AML. Data on the association of drug exposure and DAMPs are listed with references in the conventional chemotherapies paragraph (page 3, lines 75-86).

“…The role of genetic mutations in leukemia immunosurveillance should also be discussed…”

This topic has been discussed (pages 9-10, lines 441-454).

“…The authors should analyze which of the immunologic effect of the anti-leukemia drug might be a biomarker of prognosis or a target for immunotherapy…”

The text has been modified as suggested (page 11, lines 521-524).

“…The mechanism of GvL effect should be analyzed and correlated with the anti-leukemic drug associated deregulation of the immune system…”

The text has been modified as suggested (page 10, lines 458-468).

“…The term “cancer immunoediting” should be described…”

The text has been modified as suggested (page 9, lines 434-441).

Reviewer 3 Report

Tumor cells, including AML cells, have complex interactions with their microenvironment, tumor cells can remodel the microenvironment to support their proliferation and survival, escaping from the immunological surveillance. When drugs were used to treat AML patients, the AML cells serve as the targets of the drugs, but the microenvironment also has been affected. In this manuscript, Dr. Francesco Tarantini et al. try to discuss the situation in the microenvironment when drug treatment was performed. The authors introduced some widely used drugs, including chemotherapy drugs and target drugs. Overall, the manuscript was organized well and the discission is in detail, providing new thoughts on combination treatment, which may benefit the precision medicine field. The reviewer didn’t have any major concerns about the manuscript, wish the authors can add some paragraphs about the Tet2 mutant treatment related to the “4. Hypomethylating Agents” part. As epigenetic regulators, TET2 and DNMT3A/B mutants will cause the dysregulation in DNA methylation, promotes MDS and AML. Other minor issues, MLL, and AML1-ETO mutants are accounted for about 10% and 25% in adult AML patients, please discuss if possible.

Author Response

Reviewer #3:

“…The reviewer didn’t have any major concerns about the manuscript. Wish the authors can add some paragraphs about Tet2 mutant treatment related to the “4. Hypomethylating Agents” part…”

Accordingly, we added information about TET2 and epigenetic modifiers (page 5, lines 207-209).

“…Other minor issues, MLL, and AML1-ETO mutants are accounted for about 10% and 25% in adult AML patients, please discuss if possible...”

The text has been modified as suggested (page 10, lines 449-453).

Round 2

Reviewer 2 Report

The authors respond successfully to the comments